# The Similarities and Differences in the Localization of Buddhism and Christianity—Taking the Discussional Strategies and Intellectual Backgrounds of Tertullian's *Apology* and Mou Zi's *Answers to the Skeptics* as Examples †

Lin Wang

Department of History, Beijing Normal University, Beijing 100871, China; 202031120028@mail.bnu.edu.cn
† On the translation of Chinese Buddha classics, such as *Li Huo Lun* 理惑论 [*Answers to the Skeptics*], *Hong Ming Ji* 弘明集 [*Spreading the way and Elucidating the Teaching: A Collection of Expiditions Truth*], *Chu Sangzang Ji Ji* 出三藏记集 [*Catalogue of Tripitaka Works Translated into Chinese*], *Shamen Bu Jing Wang Zhe Lun* 沙门不敬王者论 [*Controversy over Shramana's solution to the monarch*] and so on. See 陈观胜, 李培荣编, 中英佛教词典, Beijing: Foreign Languages Press, n.d.

**Abstract:** After the rise of Christianity in the Roman Empire and the introduction of Buddhism into China, Christianity and Buddhism were both faced with the adjustment of the existing society. In the Roman Empire, faced with some censure, apologists began to write articles to clarify misunderstandings and express their beliefs. At the same time, there are similar argumentative documents on Buddhism in China. Their argumentation ideas also have many similarities, such as, firstly, distinguishing them from the original ideas, then using the existing ideas, and finally, actively integrating them into existing society. However, there are some bigger differences in the background of the debate between the Roman Empire and China—Christianity has strong political independence. The most fundamental difference is the atmosphere of the existing ruling ideology—China has been Confucianized, and the political independence of Confucianism is relatively weak. It is this fundamental difference that finally led to the final difference in the development paths of Christianity in the Roman Empire and Buddhism in China, which then affected their historical paths.

**Keywords:** *Apology*; *Answers to the Skeptics* (理惑论); early Christianity and Greek philosophy; the reconciliation of Buddhism; Taoism and Confucianism

## 1. Introduction

In the process of an introduction into an alien area of emerging religions, there is bound to be conflict or collision with the original beliefs. For new religions, how to cope with conflict and form an accommodation with established beliefs or cultures is referred to by some scholars as localization. This is a very important matter, and to a large extent, the fate of a religion depends on it. There are a number of such cases in history, of which the rise of Christianity in the Roman Empire and the introduction of Buddhism to China are representative examples.

After the rise of Christianity, the religion was misunderstood and even hostile by non-Christians in the Roman Empire. In order to clarify the misunderstanding and demonstrate the doctrine, so as to avoid persecution and even further spread of Christianity, a group of apologists came into being. This group included Justin Martyr, Aristides, Tatian, Clement of Alexandria, Ignatius, Athenagoras, and Tertullian. As the representative of the guardian, Tertullian's works have the most systematic and typical debate strategy. Tertullian was a Christian writer of Carthage in North Africa during the Roman Empire, known as "the last Greek apologetist and the original Latin defender" ([Campenhhausen 1964](#)). He lived in the second half of the 2nd century and the first half of the 3rd century, and he was the son of a centurion of the Roman Empire in North Africa. Originally, he was a pagan man,

and then he converted to Christianity. He studied philosophy, literature, and law, and he practiced certain Greek cultural traditions. Tertullian wrote many apologetic works, of which the *Apology* is typical. In response to the reproaches of the pagans, Tertullian responded with a warlike rebuttal, firstly to demonstrate Christian political loyalty to the empire, secondly to emphasize that Christianity was not independent of Roman tradition and society, and thirdly to demonstrate that Christianity was also a religion of reason and not superstition. He did this to reduce the resistance to the transmission of Christianity and to be better able to preach. The choice of such a debate strategy and topic was based on the arrangement of the previous apologetics documents, as well as their thematic content and debate strategies.

Similarly, when Buddhism came to China, there were many criticisms and diatribes. For the same purpose, that is, to protect and propagate the religion, the Buddhist monk Sang Yo (僧佑) compiled *Spreading the Way and Elucidating the Teaching: A Collection of Expeditions of Truth* (弘明集), the first of which is the *Answers to the Skeptics* (牟子理惑论). It is generally believed that the author of *Answers to the Skeptics* is *Mou Zi* and that it was written in the time of Liu Song Ming Emperor (刘宋明帝); scholar Lu Cheng's (陆澄) *On the Teachings of the Buddha* (法论) also states the following: "the life of Cangwu Taishou Mouzi Bo" (苍梧太守牟子博传), but no specific name is given. *Mou Zi*[1] lived in the late Eastern Han dynasty and had taken refuge in Jiaozhou (交州). He later married in Cangwu, and because of the chaos of the world, he was repeatedly hard to recruit. *Answers to the Skeptics* focuses on the history, personalities, doctrines, and teachings of Buddhism, and it uses Confucian and Taoist concepts to express Buddhism's understanding of its connection to Chinese culture, thus endeavoring to argue for the sanctity, advancement, and reasonableness of Buddhism. In the genre book, the article content adopts the form of questions and answers, which had a certain influence on the genre. In addition, the logic and form of the debate and the main topics of *Spreading the Way and Elucidating the Teaching: A Collection of Expeditions of Truth* generally follow *Answers to the Skeptics*. Therefore, it can represent the process of cultural adjustment when Buddhism was integrated into Chinese culture but did not lose its own characteristics.

Therefore, *Apology* and *Answers to the Skeptics* do not merely reflect the development dilemma and breaking process of foreign cultures after entering a strange cultural circle; instead, these works represent some situations of the conflict and integration of new cultures in communication with existing cultural thoughts, and they also represent the universality of the development of Chinese and Western cultures.

In China, there are some papers on the cultural background of Tertullian, as well as his views on death, belief, soul, and apologetic traditions (Zhang 2001; Lin and Chen 2011; Xu 2003; Wang 2001; Qi 2020); but in the Western sphere, scholars explore the status of Tertullian from the aspects of apologetics' historical traditions, doctrines, literature, influences, and localization of rhetoric (Merrill 1918; Lehmann 1959; Keresztes 1966; Barnes 1971; Sider 1971, 2001; Burrows 1988; Rives 1994; Rankin 1995; Osborn 2001; Dunn 2004; Wilhite 2007; Willert 2014). At the same time, Chinese scholars have also conducted a detailed analysis of *Answers to the Skeptics* from the aspects of the authenticity, edition, age of writing, author, and name changes[2]; there are also related discussions on the reflection of Confucian and Taoist thoughts and the process and characteristics of the Sinicization of Buddhism (Liu 2007; Mao and Tang 2013; Wang 2015; Wu 2016). There is still room for further research from the perspective of debate strategy. Compared with China, Western scholars have published less research on *Answers to the Skeptics*.[3] In collecting source materials, the author of this article did not find a comparative study of the similarities and differences between Tertullian's *Apology* and *Mou Zi*'s *Answers to the Skeptics*.[4] Therefore, this article tries to compare the similarities and differences between the two arguments through examples in order to understand the differences in religious thought between China and the West.

## 2. Divide First and Then Unite—Similarities between the Argumentative Strategies of *Apology* and *Answers to the Skeptics*

### 2.1. The Distinction from the Pre-Existing Thought

At the beginning of Christianity, the people of Rome could not accurately distinguish between Christianity and Judaism. So in order for Christianity to develop, the apologists had to distinguish themselves as separate from Judaism after initially utilizing it. This is the logical starting point of *Apology*.

"The Jewish people has deviated the path of God" is the basic starting point and foothold of the Christians' comments on the Jews; Christianity, in order to be independent and develop, had to break away from the yoke of Judaism, or be cut off from Judaism, and define its own characteristics in order to not be confused with Judaism. In *Apology*, first, Tertullian emphasizes "nor do we have any different thoughts of God" (Tertullian 1950, p. 61). Immediately after speaking of the Jews' former glory, the pen turns and writes that they are in "sin" because they "refuse to admit" their mistakes, and this is because they believe that the savior has not come, which leads to their great dispersion (Tertullian 1950, pp. 61–62). And Christians believe that Christ is the savior, the god, the Lord, master, and "the Enlightener and Guide of the human race" (Tertullian 1950, p. 62) because "with a word He drove evil spirits from men, gave sight again, to the blind, cleansed lepers, healed paralytics, and finally, by a word, restored the dead to life; He reduced to obedience the very elements of nature, calming storms, walking upon the water" (Tertullian 1950, pp. 64–65). Christ, however, was regarded by the Jews as a "charlatan" and was even killed based on this view. Therefore, whether Jesus Christ is the savior is the most fundamental difference between the two religions. Moreover, in order to further illustrate the divinity of his religion, Tertullian set the founder of Judaism as Moses; Moses is a man, and the divinity of man is not the divine nature of God. In this way, Christianity was presented as different from Judaism and superior to Judaism.

When Buddhism was introduced to China, its practice included some of the Taoist rituals, and at that time, Taoism's fairy magic was very popular, which made it difficult for the upper classes to differentiate between Taoism and Buddhism; thus, Buddhism gained trust along with Taoism. At that time, Emperor Huan of the Han dynasty and Ying, King of Chu (楚王英), respected Buddhism but did not understand the Buddhist teachings. The purpose of Buddhist worship is longevity and profit. And Buddhism often calls itself "Chinese" ("华人") under the guise of Taoism. In addition, most famous Buddhist monks hide their teachings and do not talk about miracles; rather, they speak of cause and effect to win people's hearts. In short, Buddhism and Taoism are very similar in procedure and form, and even in some concepts.

However, with the development of Buddhism, the consciousness of Buddhist independence began to strengthen; therefore, it became necessary to draw a clear line between Buddhism and Taoism. As the earliest Buddhist paper in China, *Answers to the Skeptics* undertook this important task. As indicated by *Mou Zi*, "Buddha's words of awareness" and "the Tao's words of nothingness" (Seng 2013, pp. 14–15) illustrate the fundamental differences between the two religions in concepts and words. And later on, *Mou Zi* wrote that "Buddhists take medicine and acupuncture even when they're sick" (Seng 2013, p. 54). This is another distinction from the Taoists. Moreover, *Mou Zi* said "The Buddhists say everyone dies. No one can avoid it" (Seng 2013, p. 60). Even Yao (尧), Shun (舜), Yu (禹), Bo Yi (伯夷), Shu Qi (叔齐), Zhou Wen Wang (周文王), Wu Wang (周武王), Zhou Gong (周公), Confucius, and so on, did not live to be a hundred years old, and Zi Lu (子路), Zeng Sen (曾参), and Yan Yuan (颜渊) talked about life having an end (Seng 2013, p. 61). This is different from what Taoism says: "Yao, Shun, Zhou Gong, and Confucius, and the seventy-two disciples of Confucius, all immortalized immortality" (Seng 2013, p. 60). *Mou Zi* also considered these statements to be "demonic and delusional words". As for daily life, Buddhists abstain from wine and meat and eat grains, unlike Taoists who do not eat grains but consume wine and meat (Seng 2013, p. 52). Someone asked the following question: "Wang Qiao (王乔) and Chi Songzi (赤松子) both became immortals and wrote a 170-volume book,

which tells of the ways in which immortals live forever, is this the same as what the Buddhist scriptures say?" The answer was as follows: "To compare these books with the Buddhist scriptures is, in terms of quality, like comparing the five hegemonic lords of the time of Chun Qiu (春秋五霸) with rulers of Chinese remote antiquity (三皇五帝), or the Yang Huo (阳货) of the state of Lu in the time of Chun Qiu with Confucius; in terms of form, it is like comparing a small hill with a high mountain, or a small stream with a great river and a great sea; and in terms of literary excellence, it is like comparing a tiger's skin without hair with a sheep's skin, or coarse clothing and linen with brocade. In terms of honor and generosity, despite Taoism has ninety-six spells, it can not still match for the Buddhist spells. Although the Taoist book is written in many words and sprawling, one can not get the gist of it, that's why Buddhists think they don not work. That's why you don't use this method and despise it. How can these two types of books have the same impact and effect?" (Seng 2013, pp. 50–51).

### 2.2. The Use of the Pre-Existing Ideas

An important reason why two originally different cultures can eventually intermingle is that one culture adapts by adjusting itself to the other, thus becoming similar or comparable, but not identical. This exchange and diffusion between heterogeneous cultures is the driving force behind cultural development because the exchange and diffusion of cultures inevitably lead to the expansion of cultural systems and contact with heterogeneous cultures. Moreover, there is an export and import of cultures, which propels the development of cultures. This is called "inclusion". This kind of inclusion is embodied in the use of original ideas and the active integration of ideas into society.

If Tertullian's former argument was more about clarifying misunderstandings and seeking legal status, the latter two arguments were designed to integrate into Roman society and achieve spiritual dominance. In general, Tertullian uses rational logic and Greek philosophical terms, but he does not believe them; instead, he sees them as bridges to make their debate more powerful and valid.[5] The Bible is the source of faith, but its content must be argued to convince society of its merit; therefore, rational logic comes in handy. Moreover, Tertullian used methods of logic and rational argumentation very similar to the Stoics and even the Neoplatonists (Waszink 1955). Specifically, in the face of accusations against infanticide, incest, and bloodlust, Tertullian's logical starting point is to assume that Christianity is a mysterious organization that never leaks to outsiders and firmly believes in it, and this judgment is given to Christians by outside believers. If so, how do outsiders know the actions of Christians? Moreover, because Christians do not eat blood or internal organs, and do not eat animals that die by suffocation or die naturally, infanticide and sacrifice are even more nonsense than greater sins. Even reason can blame non-Christians in turn, because "How is it that, when you are confident that they will shudder at the blood of an animal, you believe they will pant eagerly after human blood? Is it, perchance, that you have found the latter more to your taste?" (Tertullian 1950, 9: 14, p. 33). All of these argument methods reflect the absorption and utilization of Greek philosophy by Tertullian.

In addition, in order to secure a public trial, it is against justice and logic to allow those who are truly guilty to plead without the Christians. This is also where Tertullian uses rational logic. Trajan, when debriefed by Pliny Jr., said "men of this kind should not be sought out, but, when brought to court, they should be punished" (Tertullian 1950, 2: 7, p. 11). This also seems to be inconsistent with logic to Tertullian, because punishment indicates guilt, but then it is not pursued; conversely, if it is not pursued, it indicates innocence, so why would innocence be punished? This is where the contradictions come to arise. In this way, Tertullian thinks this is playing with Roman law (Tertullian 1950, 2: 8–11, pp. 11–12).

In the treatment of the Roman gods, if there are many gods, then the creators of these gods are a problem; therefore, with the following argument, Tertullian confirmed that the polygods of Rome are not worthy of respect because they are the creature, not the creator, that is, not the original one, and the original one is God. He went on to point to the artifi-

cial gods, stating that the king was above the gods, but still below God. In fact, the logical starting point of Tertullian is that Christians are people who believe and have a high moral level. Only by convincing people of Christianity, can Greek philosophy and logical reason work. Therefore, we can see that Tertullian has used the logical reasoning of Greek philosophy everywhere to demonstrate the existence of God and the rationality of the social order at that time. That is to say, in Tertullian's view, sensibility is lower than the reason purpose of Greek philosophy, which is to serve faith. This "emphasizes the role of reason on the one hand and the fundamental role of faith; the recognition of classical philosophy and the transcendence of Christianity" (Wu 2018). Therefore, it can be said that Tertullian's Christianity is "Platonized Christianity" and goes beyond Platonism.

Liang Qichao's remarks on the convergence of Confucianism and Buddhism during the Han and Wei dynasties stated that "The starting point of Sakyamuni's teachings is that there is no fathers and no rulers, which is in conflict with all the traditional doctrines and political systems of our country. After importation, if it remains unchanged for a long time, it will certainly be difficult to preserve" (Liang 2001). Tang Yongtong also said, "Since the rise of the Han Dynasty, Confucianism has been the mainstay of Chinese academia and the middle and lower regions of the Yellow river of China (中原) has been the mainstay of Chinese culture. In the beginning, foreign religions were still attached to the prophecy of yin and yang (阴阳), in order to fight for their own place among the many religious cultures." In the first chapter of *Spreading the Way and Elucidating the Teaching: A Collection of Expeditions of Truth*, that is, *Answers to the Skeptics*, the trend of the blending of Confucianism and Buddhism has emerged. *Answers to the Skeptics* cited Confucius nine times, which shows how familiar the author was with the Confucian classics. As a young man, *Mou Zi* "read extensively the classical works of Confucianism and the books of other schools, both long and short, and there was nothing he would not read" (Seng 2013, p. 6). This shows that Buddhism was in deep communication with Confucianism at a very early stage. Buddhism's first task was to define itself before one could argue that Buddhism was not in conflict with Confucianism. *Mou Zi* said that "Buddha is an honorific title for the dead, just like rulers of Chinese remote antiquity" (Seng 2013, p. 14). Furthermore, *Mou Zi* found similarities between the Buddhist doctrine of reincarnation and the Confucian doctrine of the immortality of the soul and the wheel of retribution: "Q. The Buddha said, 'A man should rise again from the dead.' I do not believe that this is true either. *Mou Zi* answers: A man has just died and his family returns to the roof to shout his name. The man was already dead, who was he calling out to in this way? Someone said: Calling out his soul. *Mou Zi* asked: If the spirit and soul of a man come back, the man will come back to life; if it does not come back, then where will it go? Said: Then it will become ghosts and gods. *Mou Zi* said: It is so. The spirit and soul of man do not die, only the body decays. The body is like the roots and leaves of the grains, and the spirit is like the seeds of the grains; the roots and leaves must have grown and withered, but the seeds do not become extinct, and so it is with a man who has attained the Way; though the body dies, the spirit lives on forever" (Seng 2013, p. 27). *Mou Zi* also refuted the Taoist saying of immortality by using the births and deaths of Yao, Shun, Yu, Bo Yi, Shu Qi, Zhou Wen Wang, Zhou Wu Wang, Zhou Gong, and Confucius, Zi Lu, Zeng Can, and Yan Yuan (Seng 2013, pp. 60–61). When asked rhetorically if he could not say anything about Buddhist service (佛事) without having been to the Western regions, *Mou Zi* again used the examples of Confucius, Yan Yuan, and Zi Gong to illustrate that the shadows of things can be observed even though the shapes of the things have not been seen (Seng 2013, pp. 56–57). Finally, the reproach of others also involved *Mou Zi*'s utilization of Confucian thought. Somebody asked: "You once said that the Buddhist scriptures are as vast as a sea of beautiful and elegant words, so why don not you use the language of the Buddhist scriptures, but instead use the *Shijing* and the *Shangshu* to answer my question? Could it be that the differences between the two are also seen as similarities?" (Seng 2013, p. 47). All these reflect the Buddhist use of Confucian classics and doctrines. By examining the doctrinal theories of Buddhism, we are able to see that Buddhism adopted a smooth

approach in order to overcome the maladaptive nature of dissemination and enriched the Buddhist classics by invoking Confucianism into Buddhism.

It was found that the integration of Confucianism and Buddhism was the way for Buddhism, as an alien religion, to gain legitimacy. Thus, in exploring this topic, it is important to note that Buddhists often defend themselves with a logic of argument that is ostensibly different from Confucianism but actually the same. But ultimately, Buddhism must integrate with Confucianists, which is reflected in reconciliation with the secular world.

*2.3. Actively Integrate into the Pre-Existing Society*

Tertullian not only defended Christians from legal charges but also presented a positive argument against the Roman indictment—an emphasis on the constructive role of Christianity in and for Roman society. As a member of Roman society, Tertullian was not opposed to taxes and advocated taxation for "good faith" (Tertullian 1950, 42: 8–9, pp. 107–8). He also mentioned that "we disdain no fruit of His works" (Tertullian 1950, 42: 2, p. 106) and live with the heretics. With regard to his assembly, he argued that it was not a "gang", but a mutual aid group that cared for widows, widowers, orphans, the infirm, and the sick, so that the basic order of life, such as funerals and education, could be maintained (Tertullian 1950, 39: 6, p. 99). As for the loss of his own religion, he hoped to succeed by practicing pious prayer (Tertullian 1950, 43: 2, p. 108). Therefore, he did not stubbornly oppose and flee passively to Roman society and culture, but entered into society and prompted it to change. In addition to refuting the charges against Christianity, he also volunteered, arguing that many people who looked loyal to the emperor were traitors to murder the emperor, and not one of these rebels was a Christian (Tertullian 1950, 35: 8–13, pp. 92–93).

And among the objects of his prayers, the emperor was at the top of the list. He wrote "For, in our case, we pray for the welfare of the emperors to the eternal God, the true God, the living God," and "We are the same toward the emperors as we are toward our neighbors. For, to desire evil, to do evil, to speak evil, to think evil of anyone—all are equally forbidden to us. Whatever we may not do to the emperor, we may not do to anyone else" (Tertullian 1950, 31: 1, p. 85; 36: 4, p. 94). He also mentioned that "We ask for them long life, undisturbed power, security at home, brave armies, a faithful Senate, an upright people, a peaceful world, and everything for which a man or a Caesar prays (Tertullian 1950, 30: 4, p. 86)." Tertullian even thought of the continuation of the empire, and the Christians were praying (Tertullian 1950, 32: 3, p. 88). And in contrast with the people who seemed loyal to the emperor, Tertullian said more firmly that every day, the Emperor is blessed with dignity and holiness; we Christians have more reason to say that the Emperor is our emperor because he is our God's sent, and we care more about him; and "there is one kind of concern shown in uneasiness about one's family; another, about one's enslavement" (Tertullian 1950, 35: 1, p. 90; 33: 1–2, pp. 88–89; 35: 13, p. 93). In addition, Christians are free from rebellion and regicide.

"On Destroying three important Things" (三破论) refers to "destroying one's body", (破身) "family", (破家) and "country" (破国). This is an attack by the scholars of that time on the behavior of Buddhism, which was different from that of the Chinese tradition. In ancient China, *Xiao Jing* says, "The body, hair and skin, received by the parents, do not dare to destroy," and loyalty to the king and love of the country, filial piety, and respect for elders are also daily ethical requirements, and marrying a wife and having children is also a necessary path in life. Therefore, China is a country that pays great attention to blood relations and human morality. However, Buddhists still were leaving their homes without honoring the king and without paying homage to their parents, did not marry and have no heirs, and met their friends without the appropriate etiquette, which was a great shock to the existing social order. Against this onslaught, Buddhism defends itself from the dichotomy between form and substance. Specifically, although Buddhism advocates shaving one's hair, it is considered to be a trivial matter: "Wives, children, and property, are not worth holding on to" (Seng 2013, p. 23); these are mere forms, and not detrimental

to the substance, so long as the motives are noble. He also discussed in detail the examples of someone in Qi (齐国) who grabbed his father's hair to save his life, Confucius who praised Tai Bo (泰伯) for cutting his hair, Yu Jean (豫让) who swallowed charcoal and became mute, Nie Zheng (聂政) who destroyed his face, Bo Ji (伯姬) who was burned to death because she kept her chastity rather than leave the room where the fire started, and Gao Xing (高行), a widow, who destroyed her face and defended herself in the face of a forced marriage to illustrate that the behaviors of Buddhists do not go against the norms of society (Seng 2013, p. 21). The article goes on to say, "If one has high morals, one can refrain from dwelling on the trivial matters of life. Shramana (沙门) can be said to be humble to the core by abandoning their families and possessions, giving up their wives and children, and eliminating sex, so how can one say that their behavior is contrary to the words of the sages or inconsistent with filial piety" (Seng 2013, p. 21). Later, he used the criticality of the situation to decide whether or not to break with common sense and give a hand to his drowning sister-in-law (Seng 2013, p. 31). Although Buddhists are criticized for their celibacy and considered unfilial, they see it as a return to a pristine life, free from worldly temptations: "It's the most polite thing to do" (Seng 2013, p. 23). Similarly, although Confucianism emphasizes that "appearance is the first of the five things" and "clothing is the beginning of the three virtues", and has a contemptuous or even denigrating attitude towards Buddhists who "are covered with a red cloth, eat once a day, and are closed to their six passions" (Seng 2013, p. 38), "shave their hair and are covered with a red cloth, and do not kneel and worship when they see others" (Seng 2013, p. 25), Buddhism argues that this is a manifestation of the fact that they "have their own will" and do not care about indulging in worldly materialistic pleasures, but rather are content to live in poverty and contentment, as well as a manifestation of the fact that they do not attach importance to outward form, but rather focus on inward substance (Seng 2013, p. 38). At the same time, it is used to show that the behavior of Buddhists is a sign of simplicity due to the fact that people in ancient times also acted in the same way (Seng 2013, p. 25). *Mou Zi* then concludes, "One should look at the big things rather than the small things…… (so that his) father and country will be blessed, and those who have grudges against him will not be able to find a place to lay their hands on him; and he will eventually become a Buddha, and his parents and brothers will all be saved and his parents and brothers will also be liberated." There is still a rhetorical question: "If it is not filial piety, is not benevolent, which is benevolent and filial?" (Seng 2013, p. 32). This is why Buddhists do not only do good to one person or one family, but to all people, families, and nations, and the latter is true "filial piety". In response to the misunderstanding that Buddhism "destroys the family", *Mou Zi* used the fact that Bo Yi and Shu Qi died of starvation on Shouyang Mountain (首阳山) without any descendants to illustrate that marrying and having children is not an inevitable choice for people. Arguing that if the Buddhists drink and eat meat, have wives and children, make money, deliver delusional speeches, and other practices it would "destroy the country", *Mou Zi* held that Buddhism can only regulate the behavior of people's thoughts, but cannot be specific to the real-time norms of each person; in addition, these people are not real Buddhists, followed by analogies, examples, and other ways of argumentation. In short, a Buddhist "can serve his relatives at home, rule his people in his country, and rule himself independently" (Seng 2013, p. 16). *Mou Zi*'s defense strategy can still be seen in later Buddhist apologetics, such as in Hui Yuan (慧远)'s *Controversy over Shramana's solution to the monarch*, in which he divides believers into those who are at home and those who have left home, with the former having to kneel and worship the king because of their compliance with secular norms and rituals; the latter actually kneel even though they do not kneel. This is an act of grace for the world. When someone slanders a Buddhist for participating in an uprising against the ruler, the believer argues that Buddhists "lead with compassion, do not kill the faithful, do not show off their integrity, and do not steal". This means that, as soon as people touch a weapon, he or she is no longer Buddhist, let alone mob rebels.[6]

Someone mentioned that "*Mou zi* skillfully transformed the conflict of the Buddhists' unruly behavior in the eyes of the people at that time into a conflict of concepts, and then

picked up classic quotations from the Confucian and Taoist classics to explain them, so that in the end, everything that the Shamans did was sensible and reasonable and conformed to the principles of Confucianism and Taoism, and the people who questioned their behavior became irrational instead" (Tang et al. 2014). Put briefly, in the view of Buddhists, substance is higher than form, and the differences in form between Buddhism and Confucianism do not affect the similarities in substance. This multi-level, multi-angle, and all-round communication and intermingling together promote the development of Buddhism and Chinese tradition in a close-knit manner.

However, the textual similarities are only literal; it is the larger differences that ultimately led to the differences between Christianity and Buddhism in Rome and China, and the reason for these differences is the different cultural contexts in which the two existed; therefore, it is necessary to explore the question of the environment in which the two existed.

## 3. Differences between the Backgrounds of *Apology* and *Answers to the Skeptics*

### 3.1. The Difference between Christianity and Confucianism in the Relationship between Politics and Religion

Since Christianity eventually replaced polytheism within the Roman Empire and had a significant impact on subsequent historical developments, the Roman Empire is based on Christianity rather than polytheism in contextual considerations. Similarly, even though Buddhism was introduced to China and played a major role, the official school and the main dominant ideology in China was still Confucianism; therefore, in contextual considerations, China still used Confucianism as a point of reference.

Christianity has maintained its essential independence in its history. In the early days of Christianity, the religion emphasized that one should "Give back to Caesar's and to God what is God's."(*Mark*, 12: 17.) This shows that inner faith remains paramount and primary. Furthermore, Jesus was on trial: "We have found this man subverting our nation. He opposes payment of taxes to Caesar and claims to Messiah, a King."(*Luke*, 23: 2) The same statement is found in "We must obey God rather than human beings."(*Acts*, 5: 29) Although the emperor was an emperor, higher than the "gods", he was still a created being whose power came from God, so he could not be regarded as the same as God. At the same time, Jesus also said, "My Kingdom is not of this world. If it were, my servants would fight to prevent my arrest by the Jewish leader. However, now my kingdom is from another place."(*John*, 18: 36) Therefore, "Everywhere in the New Testament is a sense of distance from power" (John 1996).

However, Buddhism entered China ruled by Confucianism, so the relationship between Confucianism and politics inevitably provided a large ideological environment. Confucianism was closely associated with politics from the very beginning of its existence: "Those who are in politics should act with fairness and justice" and "You want to do good and the people will do good." *Lun Yu* (论语) also said, "To govern with virtue is like the North Star, which resides in its place and is arched by all the stars" and "With the decree to govern the people, with the criminal law to straighten them out, the people only want to be able to avoid being punished for their crimes, but there is no sense of shame; with the moral guidance of the people, with the rites of assimilation, the people will not only have the sense of shame, but also have the heart of return to the service."(*Lun Yu, Wei Zheng* 论语·为政) These few sentences centered on the political views of Confucius. Therefore, there is no doubt that Confucius' doctrine is more strongly bonded to politics. In addition, Mencius said of the doctrine of "benevolent government" that "Those who practise benevolent government have no enemies. Your Majesty, please do not be in doubt!"(*Mencius, Part two of Liang Huiwang* (孟子·梁惠王上) as "He who relies on force and pretends to be benevolent can be a hegemon……he who relies on morality and enforces benevolence can be a king."(*Mencius, Part one of Gongsun Chou* 孟子·公孙丑上) Xun Zi, who followed in his footsteps, also said the following: "Rites and Laws should be applied in parallel" and that "the king and the hegemony should be used in parallel"; "The monarch who honours pro-

priety and justice and respects the virtuous will be king; the monarch who values the law and loves the people will be hegemonic; the monarch who loves profit and often engages in fraud will be dangerous; and the monarch who plays with power and intrigue, tilts against and frames people, and is sinister will perish"(*Xun Zi, Da Lüe* 荀子·大略); "The countries in heaven that follow the rituals are well governed, and those that do not follow the rituals are in chaos; those that follow the rituals are stable, and those that do not follow the rituals are in danger; those that follow the rituals will exist, and those that do not follow the rituals will perish"(*Xun Zi, Li Lun* 荀子·礼论); "This conquest of tyrants to punish the culprits is a great political achievement. That those who kill are put to death, and those who injure are punished, is the same for all the emperors of the ages"(*Xun Zi, Zheng Lun* 荀子·正论); and "The king is the boat, and the people are the water. The water could carry the boat and overturns it. This is what is meant by this."(*Xun Zi, Fu Guo* 荀子·富国) In a word, Confucianism involves many aspects of political rule, such as rites and music education, punishment, military affairs, finance, systems, and so on. After Confucianism was established as a monopoly by the Han dynasty, those who studied Confucianism gradually became Confucian scholars, and they were more completely subjugated to the regime. Although Confucianism in China has always existed as a mainstream and official ideology, it has never formed an organization independent of the government, and it has not been able to be organized, institutionalized, or materialized. This is because the rites of sacrificing to heaven, ancestors, and Confucius were mostly participated in by the emperor, who was the representative of politics. All the scholars of the Imperial College eventually entered the governmental offices, and the work of the governmental institutions at all levels, such as helping the poor, education, charity, and the maintenance of the social order, was mainly accomplished by the officials, and so on.

### *3.2. Differences in the Existence of the Pre-Existing Thoughts*

In the Republic and early Empire, Rome pursued a policy of religious toleration, meaning that politics did not care much about differences in religious denominations and doctrines, and so was generally able to accommodate foreign religions into the Roman pantheon, such as the Greek god Zeus becoming the Roman god Jupiter, Athena becoming Minerva, and Aphrodite becoming Venus, and even Isis from Egypt and Mithras from Persia. Protectors of regions and cities were also honored; Athena always protected Athens and Althemis always protected Ephesus. After his accession to the throne, Octavian continued his policy of religious "toleration", while at the same time promoting the ancient religion of Rome, repairing temples, and restoring festivals. Most importantly, he created the cult of the Principle in the 2nd and early 3rd centuries; coins issued by cities throughout Asia Minor accurately showed the exact shape of the temples and statues of their local gods. This environment of religious tolerance gave Christianity the opportunity to spread widely. When Christianity was in its infancy, the Imperial Government was a mere spectator, neutral in the controversy between Christianity and Judaism. Even if there was persecution, it was only temporary and localized. In the late Empire, even Symmachus the Elder, whom St. Ambrose had so bitterly denigrated in the "Altar of Victory Controversy,"[7] was not opposed to privileging Christianity, but only to preserving the old traditions. So while religious diversity was the norm in the ancient Roman world, this zero-sum game of Christianity versus polytheism was a product of the bigotry of later generations of scholars and the result of focusing on believing the writings of Christian scholars. Moreover, although contemporary researchers often emphasize Christianity's absorption of both the philosophy of the Hellenistic era and the spirit of Roman religion, Christianity eventually entered the empire as an independent, state religion. It is for this reason that Christianity was able to maintain its independence even when it was established as the state religion by political rulers, even to the extent of absolutely punishing emperors, such as Ambrose, who absolutely punished the Roman Emperor Theodosius the Great for the massacre of civilians by the Roman army. At the social level, Christians and the church also undertook social relief, order maintenance, controlled sacrifice, and other political activities that orig-

inally belonged to the empire. The most important thing is that Christianity, because of its doctrines, has formed organized and physical institutions that are able to perform various functions. This is the most different point between Christianity and Confucianism, and it is this point that has had an important impact on the subsequent history.

Unlike the state of religion during the Roman period, China did not have a religion in the Western sense during the Han dynasty but had established a ruling ideology—Confucianism.[8] Whether it was Taoism, which appeared locally at the end of the Eastern Han dynasty, or Buddhism, which was imported from outside, they could only adapt to Confucianism and accept its dominance. This situation can be seen everywhere in *Answers to the Skeptics*. At the beginning of the introduction of *Mou Zi*, it is said that he "studied both the Confucian classics and the doctrines of various scholars", but also "often questioned others with the Five Classics, and none of the Taoist was able to match or over him" (Seng 2013, pp. 6–7). When asked about the title of Buddha, he said, "The Buddha is also a title, as rulers of Chinese remote antiquity." This is also an example of lecturing with the help of traditional Chinese deification. In the face of the reproach, speaking directly to the point, *Mou Zi*, based on the example of Confucius, stated that "Confucius does not think that the Five Classics have been completed, and wrote *Chun Qiu* and the *Xiao Jing*" (Seng 2013, p. 18). In response to the Buddha's thirty-two faces and eighty good qualities that are different from those of ordinary people, *Mou zi* also cites examples, such as "Yao's eyebrows have eight colors, Shun's eyes have double pupils"; "Zhou Wenwang had four breasts, Zhou Gong was hunched over"; and "Fu Xi (伏羲) had a nose like that of a dragon, the top of Confucius' (仲尼) head was high in all directions with the middle being low, and Lao Zi's forehead was high and raised, and he was born with a double bridge of the nose, and there were ten kinds of stripes on the palms of his hands and the palms of his feet" (Seng 2013, p. 20).

## 4. Conclusions

In general, any new culture will be misunderstood when they initially enter the existing cultural circle, and it is normal that they are forced to make various adjustments to relieve their discomfort. Take Christianity, for example, both Greek philosophy and Judaism are used as tools to argue that Christianity is the only correct and just religion. As Mark Burrows said, "Tertullian's 'audience,' therefore, is not only that of the 'magistrates of the Roman empire,' as he claims at the outset. Rather, he is here entering into discourse with a diverse gathering of outstanding historians, encyclopedists, and rhetors of antiquity, including Varro and Tacitus, Cicero and Quintilian, and, of course, Josephus" (Burrows 1988). Whether he was fiercely discerning the differences in Judaism or using the rational logic of Greek philosophical discourse, his sole purpose was to argue for the justification of the existence of Christianity. This is also due to the fact that Christianity was in a growth period and did not have much power to completely deviate from the existing network of ideas. The same is true in *Answers to the Skeptics*. Although the author opposed Taoism and challenged the position of Confucianism, the confluence of the three religions in the Tang dynasty was manifested in respecting the king and worshiping the integration of Confucianism. The relationship between the three religions struggles but is unbroken. Meanwhile, the argumentation strategy of Buddhism demonstrated in history is to distinguish the differences from Taoism, then demonstrate its uniqueness and superiority, and finally integrate it into Confucianism.

However, the fundamental reason for the difference between the two is the different ruling ideas. To be specific, first, Confucianism is more politically dependent and less independent than Christianity. In addition, Confucianism had already been completed in the Han dynasty and was deeply involved in politics, and other ideas could only be adapted into the existing culture. Of course, Confucianism would also make certain adjustments, but its orthodoxy and subject status cannot be doubted. Second, Christianity became the state religion, and Confucianism became the "state religion" of China, which is also the background difference between Chinese culture and Occidental culture. Third, it is the

difference between Christianity and Confucianism in the understanding of the relationship between politics and religion and the existing ideological and cultural background that has had a greater effect on the historical development of religions, which finally led to the status of Christian theocracy in medieval Western Europe being higher than that of Confucianism in medieval China.

**Funding:** This research received no external funding.

**Institutional Review Board Statement:** Not applicable.

**Informed Consent Statement:** Not applicable.

**Data Availability Statement:** https://www.cnki.net/ (accessed on 1 December 2023).

**Conflicts of Interest:** The author declares no conflict of interest.

## Notes

[1] In the remaining content of *Catalogue of Tripitaka Works Translated into Chinese*, Vol. 12, that is, *the teachings of the Buddha* written by Lu Cheng of the Liusong dynasty, *Answers to the Skeptics* has this title: "the life of Cangwu Taishou Mouzi Bo", but no specific name is provided. In *Collection of catalogue of books* by *Sui shu* (隋书·经籍志), however, the author changes to "wrote by Taiwei of Han dynasty Mou Rong" ("汉太尉牟融撰"), and *Old Tang Shu* (旧唐书) and *New Tang Shu* (新唐书) follow this statement. As far as the circulation of the engraved *Spreading the Way and Elucidating the Teaching: A Collection of Expeditions of Truth* is concerned, The Ming version does not say the author is Mou Rong, but the note includes "Mou Rong of Han dynasty" ("汉牟融"), another note is "the life of Cangwu Taishou Mouzi Bo" ("一云苍梧太守牟子博传"). In the last years of the Ming dynasty, Hu Yinglin (胡应麟) pointed out that the author is not Mou Rong in *Correction of errors of Si Bu* (四部正讹); furthermore, he thought that this book is a forgery. In the Qing dynasty, Sun Xingyan (孙星衍) let his student Hong Yixuan (洪颐煊) make textual criticism, but the outcome was still uncertain, barely following the above statement. See 弘明集, 2013, 僧佑 ed. *Interposation and noted by Li Xiaorong* 李小荣笺注, 上海: 上海古籍出版社, p. 7, 笺注一; 洪颐煊. 2001. *Preface of Mou Zi* (牟子序). In *Newly Edition of unfinished Manuscript of Mou Zi* (牟子丛残新编), collected and written by 周叔迦, newly edited by 周绍良. 北京: 中国书店, pp. 73–74; *Mou Zi's Answers to the Skeptics* (牟子理惑论), 2020. Noted and translated by 梁庆寅, 北京: 东方出版社, pp. 5–7. This article does not explore this but only follows the customary use of "Mou Zi" in italics.

[2] *Mou Zi's Answers to the Skeptics* is generally regarded as *Mou Zi* (牟子); "Li Huo" ("理惑") was found in the preface. In modern times, Sun Yirang (孙诒让), Liang Qichao (梁启超), Lü Cheng (吕澂), and Chen Yuan (陈垣) all hold that this book is a forgery. Yet Liang Qichao and Lü Cheng thought this article was written between the Jin and Song dynasties, see Sun Yirang 孙诒让: Postscript of Mou Zi's Answers to the Skeptics 牟子理惑论书后, *Newly Edition of unfinished Manuscript of Mou Zi* (牟子丛残新编), pp. 75–76; Liang Qichao 梁启超: *Distinguish between true and false of Mou Zi's Answers to the Skeptics* (《牟子理惑论》辨伪), *Newly Edition of unfinished Manuscript of Mou Zi* (牟子丛残新编), pp. 77–80; Lü Cheng 吕澂: *A brief introduction to the origins of Chinese Buddhism* (中国佛学源流略讲), 北京: 中华书局, 1979, pp. 25–27. Liu Shipei 刘师培, Tang Yongtong 汤用彤, Ren Jiyu 任继愈, Yu Jiaxi 余嘉锡, Hu Shi 胡适, and Zhou Shujia 周叔迦 hold an opposing view. See Liu Shipei 刘师培: *Guo Xue Fa Wei* (国学发微), punctuate and proofread by Zhang Jinghua 张京华点校, 上海: 华东师范大学出版社, 2015, p. 57; Tang Yongtong 汤用彤: *Buddhist history of Han, Wei, Jin, Southern and Northern Dynasties(Enlarged and revised volume)* (汉魏两晋南北朝佛教史(增订本)), 北京: 北京大学出版社, 2011, pp. 71–72; Ren Jiyu 任继愈: *The History of China Buddhism* (中国佛教史) Vol. 1, 北京: 中国社会科学出版社, 1985, pp. 188–230; Yu Jiaxi 余嘉锡: Examination of Mou Zi's Answers to the Skeptics (牟子理惑论检讨), *Newly Edition of unfinished Manuscript of Mou Zi* (牟子丛残新编), pp. 115–44; Hu Shi 胡适: *Complete works of Hu Shi* (胡适全集) (第24集): Answer to Chen Yuan (答陈垣), 06. 04. 1933, pp. 157–60; Hu Shi 胡适: *Complete works of Hu Shi* (胡适全集) (第25集): Write to Zhou Yiliang (致周一良), 07. 08. 1948, pp. 350–55; Hu Shi 胡适: *The Third Letter of Hu Shi to Zhou Shujia about Mou Zi* (与周叔迦论牟子三书), *Newly Edition of unfinished Manuscript of Mou Zi* (牟子丛残新编), pp. 85–89; this also includes the opinion of Zhou Yiliang (周一良). Zhou Yiliang regards *Mou Zi's Answers to the Skeptics* as originally a Taoist work. Also see Zhou Yiliang 周一良: The Research of The Times Mou Zi's Answers to the Skeptics (《牟子理惑论》时代考), 周一良著: *Historical essays on Wei, Jin and Southern and Northern Dynasties* (魏晋南北朝史论集), 北京: 北京大学出版社, 2010, 2nd, pp. 259–71; Zhou Shujia 周叔迦: *Discussion with Liang Qichao about Mou Zi distinguishing true and false* (梁任公牟子辨伪之商榷), *Newly Edition of unfinished Manuscript of Mou Zi* (牟子丛残新编), pp. 81–84. On the version spread, name change, and recent research review, see *Spreading the Way and Elucidating the Teaching: A Collection of Expeditions of Truth* (弘明集), edited by Seng Yo 僧佑 (编): Interposition and noted by Li Xiaorong 李小荣笺注, 上海: 上海古籍出版社, 2013, pp. 6–7, interposition and note one 笺注一;

[3] 伯希和 *(Pelliot): Resaerch on Mou Zi* (牟子考), In *Newly Edition of unfinished Manuscript of Mou Zi* (牟子丛残新编), pp. 91–114; Erik Zürcher. 1972. *The Buddhist Conquest of China: The Spread and Adaptation of Buddhism in Early Medieval China*, Leiden: Brill, p. 19. The appendix at the back of *Mou Zi's Answers to the Skeptics* has the article of Fukui, Kōjun (福井康顺), see *Mou Zi's Answers to the Skeptics*, pp. 7–13, 159–81, 193–98.

4    There are some researchers who have used the comparative method, but the focus does not overlap with this paper. See Jiang Zhejie 蒋哲杰. 2012. *Cultural and linguistic activities in Wei, Jin, Six Dynasties and late Rome* (魏晋六朝与晚期罗马的文化语言活动). PHD, 华东师范大学, 上海, 中国.

5    H. Richard Niebuhr clarified the relationship between Christian belief and culture into five categories and Tertullian is classified as the opposite category between belief and culture. See H. Richard Niebuhr. 1951. *Christ and Culture*, New York: Harper & Row, pp. 45–82. However, there are also studies that show that Tertullian has a harmony between faith and reason, see also Paul Tillich. 1968. *A Complete History of Christian Thought*, New York: Harper & Row, Pt. 1, p. 38; he thinks Tertullian "has a strong rational mind".

6    See *on Correction* (正诬论), in *Spreading the Way and Elucidating the Teaching: A Collection of Expeditions of Truth* (弘明集), edited by Seng Yo 僧佑 (编), p. 85.

7    When Symmachus took over the post of Praetorian Prefect of Rome in 384, his initial action was to try to revoke an order that had been issued two years earlier by the court in Trier. In 382, Emperor Gratian attempted to curtail the privileges of the Vestal Virgins in the city of Rome; however, he only failed to do so.

8    The question of whether Chinese Confucianism is a religion originated in the problems encountered by Western missionaries during their missionary journeys to China, when Matteo Ricci (利玛窦), in order to open up the missionary field, considered Chinese Confucianism to be a religion. In modern times, firstly, it was the duo of Kang Youwei (康有为) and Liang Qichao (梁启超) who characterized Confucianism. Both of them initially considered Confucianism to be a religion, but later LIANG Qichao changed his view that Confucianism was not a religion. Neo-Confucians, and some historians, have also discussed the religiosity of Confucianism, e.g., Koo Hungming (辜鸿铭), Xiong Shili (熊十力), Hu Shih, Cai Yuanpei (蔡元培), and Qian Mu (钱穆) believe that Confucianism in China is a religion. Although Chen Duxiu believed that Confucianism was a religion, it was a religion of "indoctrination" rather than "religion". Liu Shipei, Zhang Junmai (张君劢), Zhang Dainian (张岱年), and Chen Deng (陈登) believed in principle that Confucianism was not a religion. After 1978, Ren Jiyu, Zhang Dainian, Ji Xianlin (季羡林), Lai Yonghai (赖永海), He Guanghu (何光沪), Zhang Liwen (张立文), Li Shen (李申), and Zhang Rongming (张荣明) both continue to argue that Confucianism is a religion. Some scholars, nevertheless, say this is not true: Wang Dajian (王大建) points out that "Confucianism" is a kind of magic ("儒"实为一种术数); Guo Qiyong (郭齐勇) thinks that Confucianism is a spiritual form with both humanism and religious character (儒学是既有人文主义又具备宗教性品格的精神形态); Mou Zhongjian (牟钟鉴) and Zhang Jian (张践) raise that "Confucianism" is a "patriarchal traditional religion" ("儒学"是一种"宗法性传统宗教"); and Cai Shangsi (蔡尚思) hold that Confucianism is not religious, but it plays a religious role (儒学虽非宗教但起到了宗教的作用).

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
