# Peer review of "The Similarities and Differences in the Localization of Buddhism and Christianity—Taking the Discussional Strategies and Intellectual Backgrounds of Tertullian’s Apology and Mou Zi’s Answers to the Skeptics as Examples†"

_religions, doi:10.3390/rel15010105_

Round 1
Reviewer 1 Report
Comments and Suggestions for Authors
This is an original essay that demonstrate good knowledge of important chapters in the history of Christianity and Buddhism and their initial receptions in two very different societies and empires. I read English, as well as a number of other languages and the article is well written and coherent. However, I do not read Chinese and therefore have a problem with extensive references in Chinese in an English language article. Perhaps the writers can add explanations or translations in English?
Comments on the Quality of English LanguagePlease add translations or explanations to references in Chinese. Many, if not most readers will not be able to understand the references.
Author Response
Thank you very much! If this article chould be accepted, I wil try to upload references' explanations or translations in English.
Reviewer 2 Report
Comments and Suggestions for Authors
The article aims to analyze an important and interesting topic by comparing the sources of two authors who are a central point of reference for the two religions under consideration and for the two geographical contexts of the analysis.
My concern relates to the over-generality of the arguments discussed, theses argued, and conclusions. From my point of view, it does not help that there are too long quotations in the article that are not matched by adequate discussion and analysis. For example in the conclusions the author talks about the role of the state and politics for Christianity, but I would contextualize the statements more precisely also because the term Christianity itself is broad and it is important to understand well what we are referring to. Another example is related to what is stated on p.10: what does it mean that the West is based on Christianity? What do we mean by the West? What Christianity? The same problem, I think applies to the statement at the end of p. 7 concerning the reconciliation of Confucianists with the secular world.
In addition, I would clarify in more detail and precision the choice reported in footnote 9., p. 4. and translate or transliterate the titles given in the notes on p. 3;4;5;7;9;10;11;13 so that even a reader who does not know the language can get an idea of the title being quoted.
Comments on the Quality of English LanguageI believe that the paper should be revised by a native English speaker.
Author Response
Thanks for your advice! Your suggestions are very targeted and effective. I check my article once more, and solve some problem of definetion and translation, such as "what does it mean that the West is based on Christianity? What do we mean by the West? " The west in chinese include Gansu and xinjiang Provinces and middle Asia. In Christianity, there is no suitable words.
Reviewer 3 Report
Comments and Suggestions for Authors
Please kindly expand your discussion about the interplay of Roman Empire State (Politics) and Christianity in the second paragraph of your 2.1. Doing so will make your statement-Christianity strongly politically independent more evidential.
Please kindly state out if not the exact at least approximate timespan of the Christian and Buddhist Apologetic texts sampled in your study.
Author Response
Thanks for your suggestions! Approximate timespan of the Christian and Buddhist Apologetic texts sampled were showed in introduction in second paragraph and note 3. And "the interplay of Roman Empire State (Politics) and Christianity" in the second paragraph is side quest, so there is no need to write it out in detail. Hope my reply find you well.